# An investigation of the relation between life expectancy & socioeconomic variables using path analysis for Sustainable Development Goals (SDG) in Bangladesh

**Dulal Chandra Nandi**[1]*, **Md. Farhad Hossain**[1], **Pronoy Roy**[1]*, **Mohammad Safi Ullah**[2]

**1** Department of Statistics, Comilla University, Cumilla, Bangladesh, **2** Department of Mathematics, Comilla University, Cumilla, Bangladesh

\* d.c.nandi@cou.ac.bd (DCN); pronoyroy.pt@gmail.com (PR)

**Data Availability Statement:** Supplementary data associated with this article can be found at https://data.worldbank.org/country/BD.

## Abstract

In today's world, the key variable for measuring population health is life expectancy (LE). The purpose of this research is to find out how life expectancy is related to other factors and develop a model to account for the predictors that contribute to LE. This study is also conducted to investigate and measure the effect of socioeconomic variables on LE in Bangladesh. In this study, the predictor variables are employment rate, gross national income (GNI), population growth rate, unemployment rate, and age dependency ratio. **Path analysis** disintegrated **bivariate analysis** and showed that employment rate, GNI, and age dependency ratio are significantly related to life expectancy, although bivariate analysis showed all variables are significantly related to LE. The maximum values of significant factors, GNI and employment rates, are $1930 and 21.32% happened in 2019, which is positively correlated with life expectancy. Also, the maximum value of the age dependency ratio (81.52%) happened in 1991, whereas the maximum value of the dependent variable LE (72.59 years) happened in 2019. It has been observed that LE, GNI, and employment rates all rise with one another. There exists an adverse relationship between LE and age dependency ratio. Based on comparisons with other highly developed nations, Bangladesh's GNI needs to grow faster than other significant factors to boost life expectancy. We have forecasted variables that were significantly related to LE until 2030 for the purpose of sustainable development goals, especially the 3rd goal.

## 1 Introduction

Life expectancy is an important summary measure of a population's health and well-being. Life expectancy reflects a nation's health, economic, and social conditions, and healthcare infrastructure. Statistically, LE is the average time that an individual or other creature is expected to live from the year of their birth to their current age.

It is widely used as an indicator of a country's overall development. For one to know one country's overall condition, the phenomenon of LE plays a vital role, especially in mortality as

**Funding:** The authors received no specific funding for this work.

**Competing interests:** The authors have declared that no competing interests exist.

well as in the economic sector. High-income, developed countries have seen monumental improvements in life expectancy over the 20th and 21st centuries [1, 2]. LE has constantly been the main focus of health science. The health-related predictors of LE were the prevalence of HIV, expenditures on healthcare, mortality rates, resources for healthcare, and outcomes of healthcare. Consumption of pharmaceuticals has a positive effect on LE in advanced and middle-aged people. Vegetable and fruit consumption increased by 30% and tobacco consumption decreased by at least 2 cigarettes per day, which will help to increase LE for a 40-year-old female [3]. Several production functions of health express the technical connection between health inputs and health status, where inputs of health care can be classified into three groups: social factors, natural factors, and economic factors. Many facilities of medical care, such as increasing medical staff and doctors, could reduce mortality and increase LE (life expectancy) [4, 5]. It has been proven that increasing the availability of physicians and decreasing under-nourishment and adult illiteracy help to improve LE in a country [6]. Life expectancy (LE) is linked with the mortality rate of infants and a high literacy rate. LE increased with low infant mortality rates and high literacy rates [7]. Economic and demographic factors of life expectancy (LE) were employment rate, gender, gross national income (GNI), education, and age [8–11]. Among these factors, the strongest possible determinant of LE was gross national income. In South Korea, increased GNI had a positive impact on LE [9]. The association between LE and education was significant in Sweden, Finland, Denmark, Norway [12, 13], and other European countries [14]. Similar relationships can also be seen in Brazil [15]. Recent research in Thailand concludes that older people who have higher educational qualifications and better income have greater health satisfaction and better health outcomes [16]. The inconsistency and equality of LE have serious effects on individual and aggregate human behavior because they affect human capital investment, economic growth, fertility behavior, incentives for pension benefit claims, and intergenerational transfers [17, 18].

Although socioeconomic and demographic impacts on life expectancy (LE) have already been shown in many papers [1–18], there is no such research paper found that directly focuses on the relationship between LE and socioeconomic variables, especially for Bangladesh. Therefore, we hope the current study endeavors to complete this. The principal focus of this research is to analyze the relationships between LE and other factors, and develop a model to account for the predictors that contribute to LE. This investigation would be beneficial for Bangladesh to understand which factors have the largest impact on life expectancy.

## 2 Significance of the method

In real-world data, it is hard to get the total association between variables without any statistical operation. That is why the total association between variables is measured using Pearson correlation coefficients (r). This study aims to explore the relationship of LE with other factors and develop a model to account for the predictors that contribute to LE. Path analysis is used to decompose bivariate analysis and measure several effects by investigating the link between the response variable and more than one predictor variable. With the help of model equations obtained from path analysis, one can easily measure all the magnitudes and relations between variables. After fitting the path analysis model, we have to test the goodness of fit to see how well the data fits into the model. As a result, it will guide the new researcher who wants something new for his study.

## 3 Objectives of the study

After rigorous study, it is clear that life expectancy plays a vital role in human health and health infrastructure development. The principal purpose of this research is to investigate whether

life expectancy is related to socioeconomic variables. This will help us to detect the socio-economic determinants of life expectancy in Bangladesh. We want to estimate several effects of significant factors on LE. This will help us to understand which factors are influencing life expectancy more. After that, we determine the best-fitted model and forecast the future conditions of significant factors for the sustainable development goals of Bangladesh. This will allow us to determine whether life expectancy will increase or decrease in the future as well as how quickly Bangladesh will achieve Sustainable Development Goals, especially the 3rd goal.

## 4 Methodology

To make the analysis precise and easy, we used different types of statistical techniques and software, such as R-Studio and SPSS. Both these two statistical programs provide a plethora of basic statistical functions. SPSS and R statistical packages are used to make analyses and predictions [19]. We so used Microsoft Excel for research purposes.

### 4.1 Data description

The beginning of any meaningful and worthwhile research is its data source. The actual data focuses on real research, which can help to make decisions and plans. We collected data from world development indicators (World Bank) [20]. We initially employed six variables to show the analysis of the data. Variables' names, identification marks, and sources are given in Table 1.

Full-time students, older people, and beggars are excluded from the unemployment rate and included in the age dependency ratio. The employment rate is considered below 60 years.

### 4.2 Pearson correlation coefficient

The Pearson correlation coefficient is used to estimate the relationship between variables. It is commonly used in linear regression. Its coefficient lies between -1 and 1.

Here,

- 1 is the indicator of a strong positive relationship between variables.

- -1 is the indicator of a strong negative relation between variables.

- 0 is the indicator of no relationship between variables.

The formula for Pearson Correlation Coefficient is,

$$r = \frac{n(\sum xy) - (\sum x)(\sum y)}{\sqrt{[n\sum x^2 - (\sum x)^2][n\sum y^2 - (\sum y)^2]}}$$

**Table 1. Introductory table.**

| Variables name | Identification mark | Source |
|---|---|---|
| GNI (current US$) | $x_1$ | World Bank |
| Unemployment rate, total (% of total labor force) | $x_2$ | World Bank |
| Employment rate (% of total employment) | $x_3$ | World Bank |
| Population growth rate (annual %) | $x_4$ | World Bank |
| Age dependency ratio (% of working-age population) | $x_5$ | World Bank |
| Life expectancy at birth, total (years) | $x_6$ | World Bank |

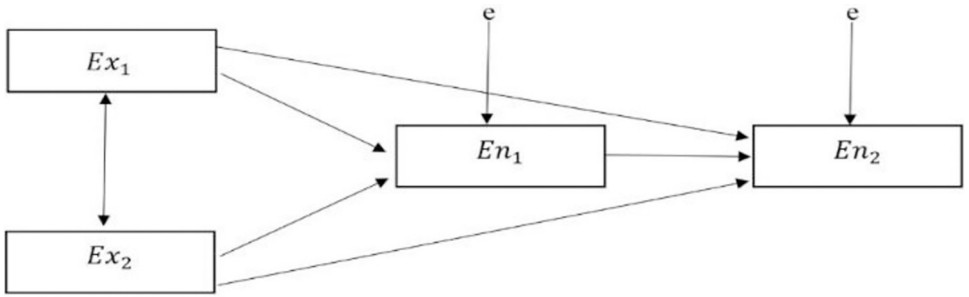

**Fig 1. Path diagram.**

### 4.3 Path analysis

A form of multiple regression analysis is Path analysis, which measures several effects models by investigating the link between the response variable and more than one predictor variable.

Fig 1 represents an example of a path diagram. Path models consist of outcome and independent variables graphically with the help of rectangle shape boxes. Variables that are not on other factors are called exogenous variables. Graphically, these variables are located at the outside edges of the model. These variables have only single-headed arrows outgoing from them. Variables that are both dependent and independent are called endogenous variables. Graphically, endogenous variables have both outgoing and ingoing arrows.

### 4.4 Stationary and non-stationary time series

The time series whose properties are not dependent on time at which it is observed is called stationary time series. On the other hand, a non-stationary series is one whose properties change over time.

### 4.5 Augmented Dickey-Fuller test (ADF)

ADF is performed to see the existence of unit roots and find out the order of integration of the variables.

### 4.6 Autoregressive Integrated Moving Average (ARIMA) process

ARIMA is a model of statistical analysis that utilizes time series data to forecast future conditions. ARIMA is a model that combines the autoregressive model $AR(p)$ with the moving average model $MA(q)$. ARIMA is formed through the lag selection of the autocorrelation function and partial autocorrelation function.

## 5 Results and discussion

### 5.1 Univariate analysis

Table 2 shows the maximum value, minimum value, mean, median, standard error of the mean, and standard deviation of study variables during the period 1991–2019. The highest life expectancy in Bangladesh was recorded in 2019 and it was 72.59 years. The maximum value of the factors, GNI, and the employment rate was also discovered in 2019, and these were $1930 and 21.32%, respectively. From the earlier data records, it is clear that gross national income, employment rate, and life expectancy all increased with one another. On the contrary, the highest values for population growth rate (2.33%) and age dependency ratio (81.52%) were recorded in 1991. In 2019, the population growth rate and age dependency ratio were both at

**Table 2. Descriptive statistics for predictor and response variables.**

|  | Mean | Median | Max Value | Min Value | Standard Deviation | Standard Error of Mean | 1st quartile | 3rd quartile |
|---|---|---|---|---|---|---|---|---|
| Unemployment rate | 3.62 | 3.91 | 5.00 | 2.20 | 0.83 | 0.15 | 2.87 | 4.30 |
| Employment rate | 14.86 | 13.79 | 21.32 | 9.78 | 3.94 | 0.73 | 11.21 | 18.56 |
| GNI | 733 | 550 | 1930 | 320 | 454 | 84 | 420 | 970 |
| Population growth rate | 1.58 | 1.48 | 2.33 | 1.03 | 0.47 | 0.09 | 1.14 | 2.08 |
| Age dependency | 63.88 | 63.05 | 81.52 | 47.92 | 10.05 | 1.87 | 55.83 | 71.78 |
| Life expectancy | 67.10 | 67.77 | 72.59 | 58.89 | 4.13 | 0.77 | 64.25 | 70.61 |

their lowest points, while the value of life expectancy was the highest. That means a high life expectancy was associated with the lowest population growth rate and the lowest age dependency ratio. The value of the unemployment rate fluctuated with time, whereas the highest rate of unemployment was 5.00% (2009).

## 5.2 Bivariate analysis

Pearson correlation coefficient is used to examine the strength and direction. It is also used to examine the linear relationship between variables.

From Table 3, it is observed that the dependent variable (life expectancy) is significantly negatively related to the population growth rate, unemployment rate, and age dependency ratio. It is significantly positively related to gross national income and employment rate. Among all major indicators of economic well-being, GNI is significantly positive in relation to employment rate and life expectancy and significantly negative in relation to age dependency ratio. The employment rate is significantly positively associated with gross national income and life expectancy. That means if the employment rate increases, then income and life expectancy will also increase. It is significant that the population growth rate is positively related to the unemployment rate, but it is negatively related to the employment rate and life expectancy. A significant negative correlation between the unemployment rate is found with the employment rate and life expectancy, and a significant positive association is found with the population growth rate. The age dependency ratio has a significant negative correlation with GNI, employment rate, and life expectancy. The age dependency ratio is found to have a positive but non-significant relationship with the population growth rate and unemployment rate. From the Pearson Correlation Coefficient for the total association, we see that all factors had a significant effect on life expectancy.

## 5.3 Path coefficient analysis

Path coefficient analysis is used here to disintegrate bivariate analysis into total effect, non-causal effect, direct effect, and indirect effect. For Path analysis, we divide our variables into two groups,

**Table 3. Pearson correlation coefficient between variables.**

|  | $x_1$ | $x_2$ | $x_3$ | $x_4$ | $x_5$ | $x_6$ |
|---|---|---|---|---|---|---|
| Gross national income ($x_1$) | 1 | -.165 | .052** | -.242 | -.274** | .436** |
| Unemployment rate ($x_2$) |  | 1 | -.398* | .857** | .037 | -.411* |
| Employment rate ($x_3$) |  |  | 1 | -.386* | -.078* | .558** |
| Population growth rate ($x_4$) |  |  |  | 1 | .017 | -.443** |
| Age dependency ratio ($x_5$) |  |  |  |  | 1 | -.393* |
| Life expectancy ($x_6$) |  |  |  |  |  | 1 |

$p^* < 0.05$ and $p^{**} < 0.01$

1. Exogenous group ($x_1$ = Gross National Income, $x_2$ = Unemployment rate, $x_3$ = Employment rate, $x_4$ = Population growth rate)

2. Endogenous group ($x_5$ = Age dependency ratio).

Here $x_6$ = Life expectancy (LE) is our dependent variable.
Linear equations for the path model are as follows.

$$x_5 = Q_{51}x_1 + Q_{52}x_2 + Q_{53}x_3 + Q_{54}x_4 + Q_{5u}R_u \tag{1}$$

$$x_6 = Q_{61}x_1 + Q_{62}x_2 + Q_{63}x_3 + Q_{64}x_4 + Q_{65}x_5 + Q_{6v}R_v \tag{2}$$

Here, Path coefficients are denoted by, $Q_{ij}$ ($i = 5,6$ and $j = 1, 2, 3, 4, 5$). $Q_{5u}R_u$ and $Q6_vR_v$ are disturbances. These disturbances are mutually independent of each other and their predictors. The residual can also be calculated from the regression equation with the help of $\sqrt{1 - R^2}$.

Path coefficient analysis of this study explores non-causal effects and total effects by counting direct and indirect effects. Path coefficients (specified in regression Eqs 1 and 2) are the direct effect of factors and are calculated by the least square regression process.

The following path models are derived from Fig 2,

$$x_5 = -0.404x_1 + 0.283x_2 - 0.205x_3 + 0.326x_4,$$

$$R^2_{5.1234} = 0.43 \tag{3}$$

$$x_6 = 0.245x_1 - 0.013x_2 + 0.386x_3 - 0.073x_4 - 0.071x_5,$$

$$R^2_{6.12345} = 0.57 \tag{4}$$

From path coefficient analysis we obtained direct effects, indirect effects, total effects, non-causal effects and the effects of these factors are given in the following table.

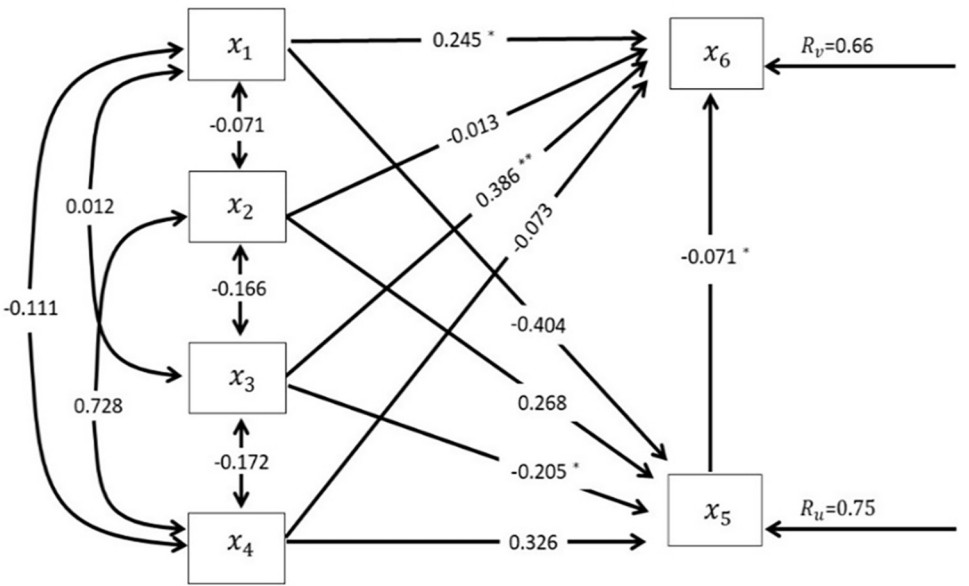

**Fig 2. Path diagram of factors affecting LE. $p^* < 0.05$ and $p^{**} < 0.01$.**

**Table 4.  Effects of independent variables on LE.**

| Endogenous variable | Exogenous variable | Total effect | Non-causal effect | Indirect effect | Direct effect | Total association |
|---|---|---|---|---|---|---|
| $x_5$ | $x_1$ | -0.404 | -0.13 | - | -0.404 | -.274** |
| | $x_2$ | 0.268 | 0.231 | - | 0.268 | .037 |
| | $x_3$ | -0.205 | -0.127 | - | -0.205* | -.078* |
| | $x_4$ | 0.326 | 0.309 | - | 0.326 | .017 |
| $x_6$ | $x_1$ | 0.27371 | -0.16229 | 0.0287 | 0.245* | .436** |
| | $x_2$ | -0.0331 | 0.3779 | -0.0201 | -0.013 | -.411* |
| | $x_3$ | 0.4013 | -0.1567 | 0.0150 | 0.386* | .558** |
| | $x_4$ | -0.0961 | 0.3469 | -0.023146 | -0.073 | -.443** |
| | $x_5$ | -0.071 | 0.322 | - | 0.071* | -.393* |

$p^* < 0.05$ and $p^{**} < 0.01$.

Calculation formula for, Total effect = Indirect effect + Direct effect.

Non-causal effect = Total effect–Total association.

From Table 4, the direct effects of GNI (0.245), employment rate (0.386) and age dependency ratio (-0.071) are significant on life expectancy (Model 4). On the other hand, only the employment rate (-0.205) has a direct significant effect on age dependency ratio (Model 3). The indirect effects of GNI (0.0287) and employment rate (0.0150) are favorable on LE through age dependency ratio, although the effect of age dependency ratio (-0.071) on life expectancy is adverse. The overall effect of employment rate (0.4013) and GNI (0.27371) is favorable on LE, but age dependency ratio (-0.017) has an adverse effect on LE. Among all the determinant predictors, path analysis showed that GNI, employment rate, and age dependency ratio have a significant role in LE. Now we have forecasted all three significant factors for the Sustainable Development Goals (SDG).

## 5.4 Univariate time series analysis

At first, we see the comparison graph of significant factors between Bangladesh and other highly developed countries (United States, Canada, Australia, United Kingdom, Norway & Denmark).

From Fig 3, all countries' employment rates, life expectancy, and age dependency ratios are almost in the same position in 2019, except for GNI. It can be said that if Bangladesh has to increase life expectancy, then GNI should increase faster than other significant factors. If it is possible to do so quickly, Bangladesh can easily increase its SDG ranking.

### 5.4.1 Checking stationarity of Gross National Income (GNI)

We have to figure out whether the GNI data is non-stationary or stationary.

Fig 4 shows the slightly downward and highly upward pattern of GNI data until 2019 in Bangladesh. This indicates that the data is not stable. The probability value of the ADF test for GNI is 0.99, which is greater than 0.05. Here, the null hypothesis is considered true, or it has a unit root. In other words, the data is non-stationary. In order to make it stationary, we need to take difference in the data.

Fig 5 shows the fourth difference time series plot. After taking the fourth difference, we get the probability value of the ADF test is 0.01, which is below 0.05. Therefore, we can accept the alternative hypothesis and say that the data is stationary.

### 5.4.2 Selection of appropriate model

The unit root test reveals significant values after taking the fourth difference. So we get the difference value d = 4. Now to choose the best ARIMA model for the data, we need to select lag

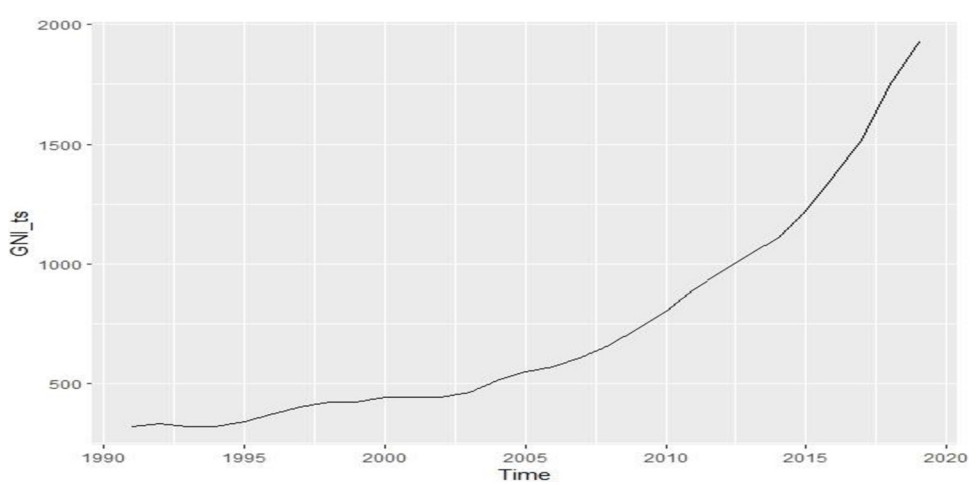

**Fig 3. Comparison graphs of factors between Bangladesh & other highly developed nations.**

**Fig 4. Time series plot of GNI.**

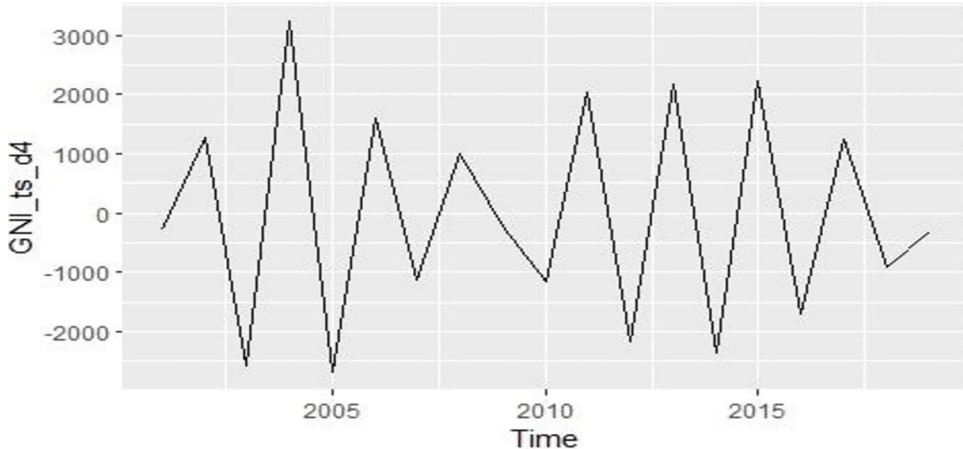

**Fig 5. Fourth difference time series plot of GNI.**

value through the plot of autocorrelation function (ACF) and partial autocorrelation function (PACF). From the PACF plot, we select lag for the autoregressive model (p), and from the ACF we select lag for the moving average model (q). The PACF and ACF plots after taking the fourth difference are given below.

It can be seen from the PACF (Fig 6) plot that the first and second spikes cross the blue-dotted significant belt. Here, the p term may be the first or second lag. From the ACF plot (Fig 6) of GNI data, lags at order first, second, third, and ninth cross the significant belt, i.e., there are four significant spikes that have crossed the confidence belt.

### 5.4.3 Checking AIC for different ordered model

Based on the PACF and ACF plots, we select the best-fitted ARIMA model. The model that gives the minimum AIC value will be considered the best ARIMA model.

From above Table 5, it is clear that *ARIMA* (1,4,2) is the best-fitted model for forecasting GNI data because it gives the minimum AIC value.

### 5.4.4 Forecasting future GNI for Bangladesh

We will forecast GNI data till 2030 for the third goal of sustainable development purpose.

Table 6 shows the future predicted values of GNI in Bangladesh. It will be increasing and the highest value will happen in 2030.

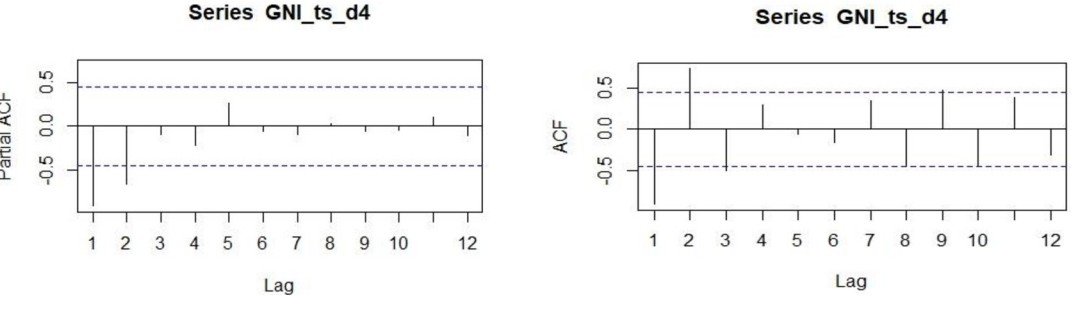

**Fig 6. PACF and ACF plot of stationary GNI.**

**Table 5. AIC values of different ordered models for forecasting GNI.**

| ARIMA (p,d,q) models | Akaike Information Criterion (AIC) |
|---|---|
| *ARIMA* (1,4,1) | 258.55 |
| *ARIMA* (1,4,2) | 254.18 |
| *ARIMA* (1,4,3) | 255.01 |
| *ARIMA* (1,4,9) | 259.56 |
| *ARIMA* (2,4,1) | 259.33 |
| . . . | . . . |
| *ARIMA* (2,4,3) | 257.97 |
| *ARIMA* (2,4,4) | 259.96 |

The future state of GNI from Table 6 is shown in Fig 7. The blue line indicates the future state of GNI. It will increase in the future. Increasing GNI is good for life expectancy (LE) because they are positively correlated. Path analysis, as well as bivariate analysis, showed that GNI was significantly positively correlated with LE in Bangladesh. From the graph, it is estimated that the LE will also increase in the future. This helps Bangladesh to achieve sustainable development goals. Here, the deep blue shaded region indicates the 80% confidence interval and the light blue shaded region indicates the 95% confidence interval.

By adopting the same process, the best-fitted model for employment rate has been selected and it is *ARIMA* (1,2,1). Based on *ARIMA* (1,2,1), the estimated future value of the employment rate until 2030 is given in the following table.

Table 7 shows the future predicted values for the employment rate in Bangladesh.

Table 7 values are represented in Fig 8. The blue line gives the future rate of employment for Bangladesh for the period 2020 to 2030. The graph represents an increasing pattern of employment rates in Bangladesh. Both path analysis and bivariate analysis revealed that the employment rate is significantly positively correlated with LE in Bangladesh. That's why increasing employment rates is good for achieving the third goal of sustainable development.

Again, for the age dependency ratio, the best-fitted model is also *ARIMA* (1,2,1). From *ARIMA* (1,2,1) we get the following estimated future value.

Table 8 shows the future value of the age dependency ratio. It will decrease in the future.

The blue line in Fig 9 indicates the future age dependency ratio status. It will continue to fall until 2030, which is good for LE. This means that the percentage of working-age people will be increasing day by day. Decreasing the age dependency ratio or increasing the

**Table 6. Forecast value for GNI using *ARIMA* (1,4,2).**

| Point | Forecast | Lo 80 | Hi 80 | Lo 95 | Hi 95 |
|---|---|---|---|---|---|
| 2020 | 2152.891 | 2118.4 | 2187.383 | 2100.141 | 2205.641 |
| 2021 | 2389.443 | 2319.276 | 2459.61 | 2282.132 | 2496.754 |
| 2022 | 2651.866 | 2533.496 | 2770.237 | 2470.835 | 2832.898 |
| 2023 | 2938.309 | 2760.256 | 3116.362 | 2666 | 3210.618 |
| 2024 | 3251.691 | 3000.892 | 3502.491 | 2868.127 | 3635.256 |
| 2025 | 3593.314 | 3255.44 | 3931.187 | 3076.581 | 4110.047 |
| 2026 | 3965.026 | 3524.09 | 4405.962 | 3290.673 | 4639.379 |
| 2027 | 4368.492 | 3806.732 | 4930.253 | 3509.354 | 5227.631 |
| 2028 | 4805.44 | 4103.153 | 5507.727 | 3731.384 | 5879.495 |
| 2029 | 5277.573 | 4412.995 | 6142.152 | 3955.314 | 6599.832 |
| 2030 | 5786.606 | 4735.789 | 6837.423 | 4179.52 | 7393.692 |

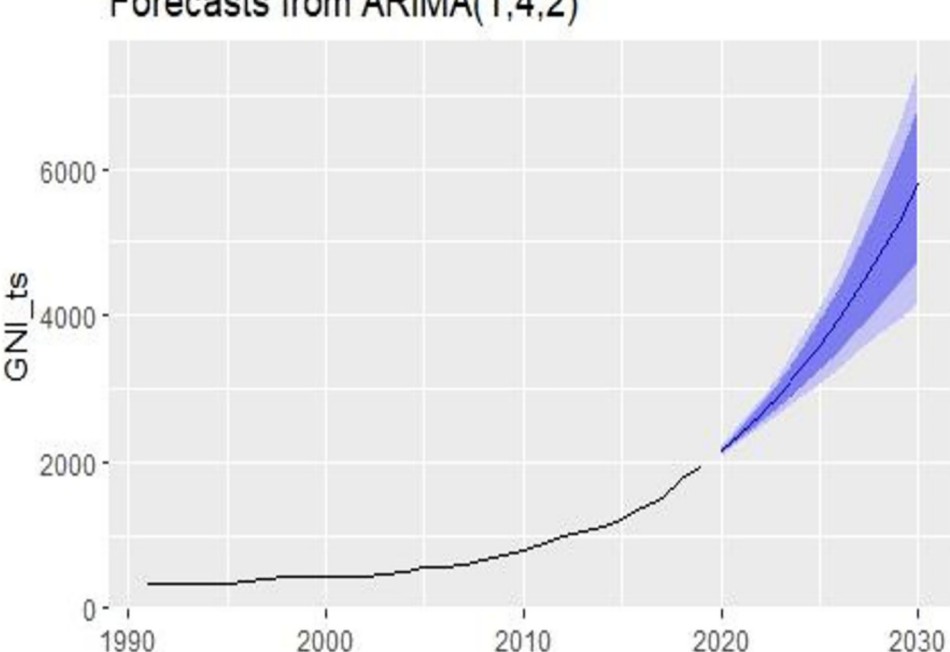

**Fig 7. Future condition of GNI in Bangladesh.**

percentage of working-aged people is good for LE, which will help Bangladesh to achieve sustainable development goals very quickly.

## 6 Conclusion

We have analyzed how factors affect life expectancy and determined predictors that could increase our life expectancy (LE). According to the analysis, GNI, employment rate, and age dependency ratio are the top determinants of LE, even though all factors have a role to play. The growth of Gross National Income and employment facilities can contribute to decreasing the age dependency ratio and increasing the LE. Based on previous data records, it has been observed that GNI, employment rate, and LE are on the rise. We checked the future value of GNI, employment rate, and age dependency ratio to see whether LE will increase or decrease.

**Table 7. Forecast value for employment rate using *ARIMA* (1,2,1).**

| Point | Forecast | Lo 80 | Hi 80 | Lo 95 | Hi 95 |
|---|---|---|---|---|---|
| 2020 | 21.84801 | 21.54733 | 22.14869 | 21.38815 | 22.30786 |
| 2021 | 22.3353 | 21.68339 | 22.9872 | 21.33829 | 23.3323 |
| 2022 | 22.85712 | 21.76548 | 23.94877 | 21.18759 | 24.52665 |
| 2023 | 23.34966 | 21.75945 | 24.93986 | 20.91765 | 25.78167 |
| 2024 | 23.86703 | 21.71432 | 26.01975 | 20.57474 | 27.15932 |
| 2025 | 24.36334 | 21.59922 | 27.12745 | 20.13599 | 28.59069 |
| 2026 | 24.87752 | 21.44999 | 28.30504 | 19.63557 | 30.11946 |
| 2027 | 25.37654 | 21.24266 | 29.51041 | 19.05432 | 31.69875 |
| 2028 | 25.88841 | 21.00346 | 30.77336 | 18.41752 | 33.3593 |
| 2029 | 26.38938 | 20.71453 | 32.06423 | 17.71045 | 35.06832 |
| 2030 | 26.8996 | 20.39511 | 33.4041 | 16.95183 | 36.84737 |

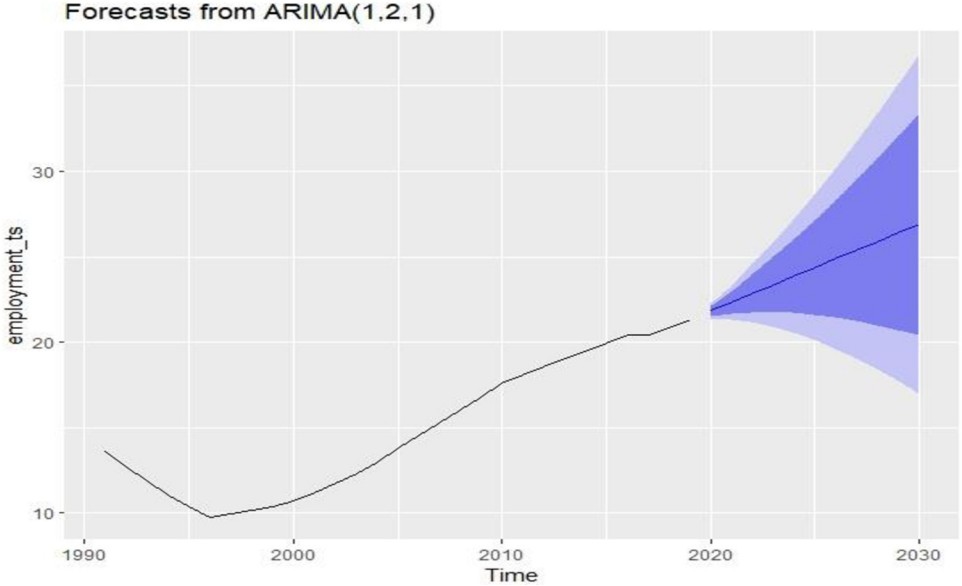

**Fig 8. Future condition of employment rate in Bangladesh.**

In conclusion, based on the time series analysis, LE is likely to increase in the upcoming days since GNI and employment rates will increase in the future while the age dependency ratio will decrease, which will help Bangladesh to achieve SDG 3rd goal very quickly.

## 7 Recommendation

The following recommendations are made to take necessary to enhance life expectancy:

- Employment rate should be increased for LE. Government should target to enhance LE by increasing job facilities in Bangladesh.

- People of this country should be self-dependent. It helps the country to increase national income and reduce the sage dependency ratio. Increasing GNI and reducing the age dependency ratio is essential to enhance average LE.

- Government should take necessary steps to control the unemployment rate, poverty and population growth to increase LE in our country.

**Table 8. Future value for age dependency ratio using *ARIMA*(1,2,1).**

| Point | Forecast | Lo 80 | Hi 80 | Lo 95 | Hi 95 |
|---|---|---|---|---|---|
| 2020 | 46.86035 | 46.72126 | 46.99945 | 46.64762 | 47.07308 |
| 2021 | 45.79149 | 45.50643 | 46.07654 | 45.35553 | 46.22744 |
| 2022 | 44.71637 | 44.27042 | 45.16231 | 44.03436 | 45.39837 |
| 2023 | 43.63793 | 43.01928 | 44.25658 | 42.69178 | 44.58407 |
| 2024 | 42.55773 | 41.75579 | 43.35966 | 41.33128 | 43.78418 |
| 2025 | 41.4766 | 40.48136 | 42.47183 | 39.95451 | 42.99868 |
| 2026 | 40.39497 | 39.19672 | 41.59321 | 38.56241 | 42.22753 |
| 2027 | 39.31307 | 37.90235 | 40.7238 | 37.15555 | 41.4706 |
| 2028 | 38.23104 | 36.59858 | 39.86351 | 35.7344 | 40.72769 |
| 2029 | 37.14894 | 35.2857 | 39.01218 | 34.29936 | 39.99852 |
| 2030 | 36.06679 | 33.96397 | 38.16962 | 32.8508 | 39.28279 |

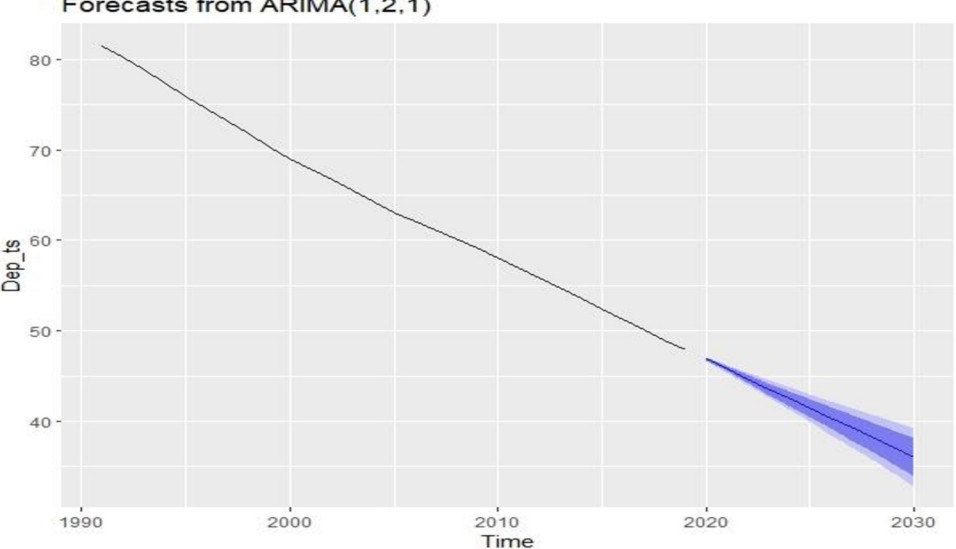

**Fig 9. Future condition of age dependency ratio in Bangladesh.**

## Author Contributions

**Conceptualization:** Dulal Chandra Nandi.

**Data curation:** Dulal Chandra Nandi, Pronoy Roy.

**Formal analysis:** Md. Farhad Hossain, Mohammad Safi Ullah.

**Methodology:** Dulal Chandra Nandi.

**Software:** Dulal Chandra Nandi, Pronoy Roy.

**Supervision:** Dulal Chandra Nandi, Md. Farhad Hossain, Mohammad Safi Ullah.

**Validation:** Dulal Chandra Nandi, Mohammad Safi Ullah.

**Writing – original draft:** Pronoy Roy.

**Writing – review & editing:** Pronoy Roy, Mohammad Safi Ullah.

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
