## [Decision Letter · Decision Letter 0]

24 Aug 2022

PONE-D-22-21852An Investigation of the Causal Relation Between Life Expectancy & Socioeconomic Variables Using Path Analysis for Sustainable Development Goals (SDG) in BangladeshPLOS ONE

Dear Dr. Dulal Chandra Nandi,

Thank you for submitting your manuscript to PLOS ONE. After careful consideration, we feel that it has merit but does not fully meet PLOS ONE’s publication criteria as it currently stands. Therefore, we invite you to submit a revised version of the manuscript that addresses the points raised during the review process.

We look forward to receiving your revised manuscript.

Kind regards,

Ricky Chee Jiun Chia

Academic Editor

PLOS ONE

Journal Requirements:

"No"

"No"

5. One of the noted authors is a group or consortium "Dulal Chandra Nandi, Md. Farhad Hossain & Pronoy Roy". In addition to naming the author group, please list the individual authors and affiliations within this group in the acknowledgments section of your manuscript. Please also indicate clearly a lead author for this group along with a contact email address.

6. We note you have included a table to which you do not refer in the text of your manuscript. Please ensure that you refer to Table e in your text; if accepted, production will need this reference to link the reader to the Table.

Reviewers' comments:

Reviewer's Responses to Questions

**Comments to the Author**

1. Is the manuscript technically sound, and do the data support the conclusions?

Reviewer #1: Partly

Reviewer #2: No

2. Has the statistical analysis been performed appropriately and rigorously? 

Reviewer #1: Yes

Reviewer #2: No

3. Have the authors made all data underlying the findings in their manuscript fully available?

Reviewer #1: No

Reviewer #2: Yes

4. Is the manuscript presented in an intelligible fashion and written in standard English?

Reviewer #1: Yes

Reviewer #2: No

5. Review Comments to the Author

Reviewer #1: The paper, in the complex, offers some interesting views and allows further considerations.

However, some aspects could be improved.

I recommend the following revisions and integrations.

First, the title mentions the concept of "causal relation" but in the text, there isn't a structured analysis of causality, the authors talk about "non-causal effect", so it is not possible to assert a causality effect of some variables on outcome variables.

In the last section, there is a bibliography, but there are no citations inside the text. Especially in the background, I recommend the presence of references included in the text. There are some sentences such as "it has been proven" or "already been shown in many papers" that without a reference are meaningless.

I consider necessary the presence of a section (maybe a table) that includes an explanation of all the variables employed and their derivation/methods of computation. This addition will considerably help the reader understand the paper's aim, the different models, and the results.

Sometimes the reading is difficult, some parts seem recurring: I suggest reporting the significant evidence resulting in the table (not all).

In the analysis of stationarity and in the research of the forecasting model, I suggest explaining in detail the procedures adopted for GNI and then trying to recap the two similar procedures employed for Employment rate and age dependency rate.

The authors should provide a numeration (also with letters if they prefer) for all the tables and figures in the text.

I recommend a check for grammar and typing.

Finally, I suggest refining the final part (the conclusion) and trying to better conclude and harmonized all the issues introduce.

Reviewer #2: The research article "An Investigation of the Causal Relation Between Life Expectancy & Socioeconomic Variables Using Path Analysis for Sustainable Development Goals (SDG) in Bangladesh" claims that its aim is to study the causal effect of employment-related variables (unemployment / employment rate, national income, population growth) on life expectancy, making policy recommendations in the context of Bangladesh (i.e., increasing employment to enhance life expectancy). However, in my view, the paper exhibits several serious weaknesses.

Contribution of the Paper and Setting

1. The paper completely lacks an Introduction which states the research questions addressed by the paper, its motivation and, most of all, its contribution to the existing literature and to the policy debate. There is a very short paragraph "Objectives of the study", but the objectives are only listed and not explained. In the present version, there is a reference list at the end of the paper, but there are no references at all in the text, so it is difficult to understand whether and how this study contributes to existing knowledge and to what extent its findings are in line or not with the literature. A brief search of literature concerning the association between employment and life expectancy seems to suggest that there is no big novelty.

2. Concerning again the contribution of the paper and its specific case study, the Background section is rather poor. A reader would expect here a brief presentation of the research setting in Bangladesh, which is the context of interest, for which the final policy conclusions are drawn. The general statements on the importance of life expectancy as an indicator, which are included in the Background, should instead be part of the Introduction which is now missing.

Data

3. As far as data are concerned, the analysis is based on the World Development Indicators provided by the World Bank. However, the description of the data is almost completely missing. There is a very short paragraph "Data description", which only lists the chosen variables, and a table of Descriptive Statistics (in the section of Results). The authors should clearly state how each variable is defined and measured, also mentioning its potential drawbacks. Beyond employment per se, whose association with life expectancy is not clear a priori, it would be interesting to consider also data on the share of people in each economic sector (or occupation), and most of all the quality of working conditions. To this purpose, there are also datasets made available by the International Labour Organization. An analysis considering more specific variables regarding the labor market would probably be more informative also from the point of view of policymakers.

In any case, most importantly, since data regard only Bangladesh, the analysis should be performed and interpreted cautiously: only one country and year-level data are not enough to have a robust and reliable analysis.

Methodology and results

4. One of the major flaws of the paper is that it is hard to believe the authors' statements of causality. The empirical analysis is not based on an identification strategy that allows to study causal relationships. On the contrary, the authors only present a statistical analysis that only suggests that there is association between life expectancy and employment / income variables. Any potential issues of reverse causality (i.e., higher life expectancy implying better quality of life leads to higher income and employment rates, and not the opposite) are not addressed and cannot be excluded. In this sense, the conclusions drawn by the authors in terms of causality are not supported by the performed analysis. Indeed, the results obtained from path analysis can be interpreted in a causal way if the authors are sure a priori that relationships between variables go in one direction only (reverse causality excluded a priori), generally because a precise time-ordering.

5. When presenting results, descriptive statistics are commented in detail (the table would be enough), while little space is devoted to the (presumed) causal relationships of interest. The numerous sections on results (including stationarity checks) are presented in a rather confused way.

General Comments on the Paper

6. The paper does not have a clear structure and it is quite confusing. While important sections are missing (e.g., Introduction), there are too many short sections (e.g., one for each stationarity check). The paper should have some main sections (Intro, Background, Data, Methodology, Results, Discussion/Conclusion) and, in case, some sub-paragraphs. The presentation of the contents of the article does not have a clear flow. There are also several typos, missing words, and some sentences which syntax problems, which in some points compromise the fluency of the paper.

7. The conclusions are very short and, while briefly summarizing the results, there is no discussion and no reference to all the potential limitations of the analysis. Despite one paragraph is titled "Results and Discussion", there is no proper discussion of the results (and of the chosen methodology) at any point.

6. PLOS authors have the option to publish the peer review history of their article (what does this mean?). If published, this will include your full peer review and any attached files.

Reviewer #1: No

Reviewer #2: No

---

## [Editor Report · Decision Letter 1]

19 Sep 2022

An Investigation of the Relation Between Life Expectancy & Socioeconomic Variables Using Path Analysis for Sustainable Development Goals (SDG) in Bangladesh.

PONE-D-22-21852R1

Dear Dr. Dulal Chandra Nandi,

We’re pleased to inform you that your manuscript has been judged scientifically suitable for publication and will be formally accepted for publication once it meets all outstanding technical requirements.

Kind regards,

Ricky Chee Jiun Chia

Academic Editor

PLOS ONE
---

## [Editor Report · Acceptance letter]

21 Sep 2022

PONE-D-22-21852R1 

An Investigation of the Relation Between Life Expectancy & Socioeconomic Variables Using Path Analysis for Sustainable Development Goals (SDG) in Bangladesh. 

Dear Dr. Nandi:

I'm pleased to inform you that your manuscript has been deemed suitable for publication in PLOS ONE. Congratulations! Your manuscript is now with our production department. 

Kind regards, 

on behalf of

Dr. Ricky Chee Jiun Chia 

Academic Editor

PLOS ONE